# A Markovian analysis of bacterial genome sequence constraints

Aaron D. Skewes[1,2] and Roy D. Welch[1]

[1] Department of Biology, Syracuse University, Syracuse, NY, United States
[2] Department of Mathematics, Syracuse University, Syracuse, NY, United States

## ABSTRACT

The arrangement of nucleotides within a bacterial chromosome is influenced by numerous factors. The degeneracy of the third codon within each reading frame allows some flexibility of nucleotide selection; however, the third nucleotide in the triplet of each codon is at least partly determined by the preceding two. This is most evident in organisms with a strong $G + C$ bias, as the degenerate codon must contribute disproportionately to maintaining that bias. Therefore, a correlation exists between the first two nucleotides and the third in all open reading frames. If the arrangement of nucleotides in a bacterial chromosome is represented as a Markov process, we would expect that the correlation would be completely captured by a second-order Markov model and an increase in the order of the model (e.g., third-, fourth-...order) would not capture any additional uncertainty in the process. In this manuscript, we present the results of a comprehensive study of the Markov property that exists in the DNA sequences of 906 bacterial chromosomes. All of the 906 bacterial chromosomes studied exhibit a statistically significant Markov property that extends beyond second-order, and therefore cannot be fully explained by codon usage. An unrooted tree containing all 906 bacterial chromosomes based on their transition probability matrices of third-order shares ~25% similarity to a tree based on sequence homologies of 16S rRNA sequences. This congruence to the 16S rRNA tree is greater than for trees based on lower-order models (e.g., second-order), and higher-order models result in diminishing improvements in congruence. A nucleotide correlation most likely exists within every bacterial chromosome that extends past three nucleotides. This correlation places significant limits on the number of nucleotide sequences that can represent probable bacterial chromosomes. Transition matrix usage is largely conserved by taxa, indicating that this property is likely inherited, however some important exceptions exist that may indicate the convergent evolution of some bacteria.

## INTRODUCTION

For more than twenty years, the nucleotide composition of bacterial genomes has been the focus of many studies attempting to identify patterns in nucleic acid sequences. One of the first analyses of nucleotide sequences by Muto and Osawa noted that nucleotide

Corresponding author
Roy D. Welch, rowelch@syr.edu

biases exist and are likely influenced by selection (*Muto & Osawa, 1987*). Later work by Kariin and Burge proposed that a bacterial signature could be defined by certain statistical properties of complete sequences (*Kariin & Burge, 1995*). They discovered that correlations exist between neighboring nucleotides (dinucleotides) in bacteria, and that dinucleotide frequencies can be used as a genomic signature which may result from: (1) the chemistry of dinucleotide stacking; (2) DNA conformational tendencies; (3) species-specific properties of DNA replication and repair mechanisms; (4) the selection of restriction endonucleases (*Karlin, Campbell & Mrázek, 1998*); and (5) codon usage, as it effects translational efficiency (*Gouy & Gautier, 1982*; *Grantham et al., 1981*; *Sharp et al., 1993*). These and other pioneering studies were narrow in scope because, at that time, available data was limited to single gene sequences, partial chromosomes, and the complete genomes of a small number of model organisms, such as *Escherichia coli K-12* (*Blattner et al., 1997*), *Haemophilus influenzae* (*Fleischmann et al., 1995*) and *Bacillus subtilis* (*Kunst et al., 1997*). Nevertheless, these analyses were instrumental in laying the foundation for statistical genomics. In this early period, researchers were forced to focus on very specific phenomena or draw broad conclusions from data sets that were insignificant when compared to the size of the global metagenome. The situation is beginning to change. Genome sequences are now available for more than 2,000 bacterial species, which may represent as much as ∼0.002% of all bacteria (*Curtis, Sloan & Scannell, 2002*; *Schloss & Handelsman, 2004*). With this expanded data set we can begin to address new types of questions. For example, we can begin to identify sequence features that may constrain nearly all bacterial genomes, and thereby describe a set of heuristics that may eventually help define the statistical boundaries of what constitutes a bacterium.

One established method used to model genome sequences is the finite state Markov chain model (see Methods) (*Almagor, 1983*; *Avery, 1987*; *Blaisdell, 1985*; *Brendel, Beckmann & Trifonov, 1986*; *Gelfand, Kozhukhin & Pevzner, 1992*). Markov models are defined by a transition matrix, which stores the conditional probabilities, in the case of a finite sequence, of the $k$th symbol following the previous $k-1$ symbols in a word of length $k$; they are akin to word frequency counts. A Markov chain model considers the transitions to be a stochastic process, defined by the conditional probabilities of each transition. The conditional probabilities can either be estimated or calculated precisely based on the sequence, as is our case. This differs from a frequency analysis in an important way; the transition probabilities are conditional, representing the probability of the transition given the previous states (previous $k-1$ nucleotides) and so it is not a measure of the frequency for a particular sequence. Applying this type of analysis to a complete genome sequence provides information about dynamic and stationary statistics that cannot be captured from a single gene or set of genes. One of the first applications of Markov models to the analysis of genetic sequences was their use as a method to identify sequence bias. Pioneering work by researchers including Phillips (*Phillips, Arnold & Ivarie, 1987*), Rocha (*Rocha, Viari & Danchin, 1998*), and Burge and Karlin (*Burge, Campbell & Karlin, 1992*) established that Markov analysis of DNA sequences can be useful in identifying over- and under-represented sequences. Work by *Elhai (2001)* compared several different statistical

methods of finding bias in the relative abundance of oligonucleotides in DNA sequences. All these methods were based on comparing observed oligonucleotide frequencies to their expectation under several models, and all concluded that Markov model based methods underperformed some more complex methods, when the purpose of the method was to determine abundance.

Determining relative abundance is not the only reason for examining DNA sequences, however, and when looking for other patterns an empirically derived Markov model does contain valuable information. For example, lateral gene transfer events produce a localized nucleotide bias that can be detected with variations of Markov models, although they must be recent events, as the bias tends to disappear in a short period of evolutionary time (*Lawrence & Ochman, 1997*; *Reva & Tummler, 2004*). Many studies have examined this phenomenon and have concluded that the lateral transfer of genetic material is a very important factor in bacterial evolution (*Campbell, 2000*; *Doolittle, 1999*; *Jain, Rivera & Lake, 1999*; *Koonin, Makarova & Aravind, 2001*; *Woese, 1998*). This conclusion may seem obvious now but, at the time, it challenged many assumptions about vertical descent, the meaning of phylogenetics, and how phylogenies are constructed (*Ludwig & Klenk, 2005*). This and similar Markov model based methods have also revealed niche and habitat influences in the genomic composition of bacteria at the $G + C$ content level (*Foerstner et al., 2005*), the amino-acid level (*Suen, Goldman & Welch, 2007*), and the whole genome level (*Perry & Beiko, 2010*). One of the earliest gene prediction methods (*Borodovsky & McIninch, 1993*) used non-homogenous Markov models to estimate the probability that a particular location along the genome of an organism contains genetic information. Clearly, Markov models have a purpose in evaluating patterns in DNA sequences, although they may not always be the best choice.

Many studies have explored the use of Markov models to infer phylogenies, as an alternative to methods based on multiple sequence alignment. *Höhl & Ragan (2007)* compared several alignment-free methods for inference of phylogeny based on bacterial amino acid sequences. They made two important conclusions: (1) methods based on k-mer frequencies are generally inferior to approaches based on maximum-likelihood distance estimates of multiply aligned sequences and; (2) there is an optimal word length ($k$) which produces a stable inferred tree, beyond which there is only a negligible improvement in stability. A similar conclusion was reached by *Jun et al. (2010)* and *Dai & Wang (2008)* using proteome sequences of prokaryotes. Again, it must be noted that these studies were looking for optimal ways to identify a particular set of data; their conclusions do not mean that Markov methods are inherently inferior. There is a significant amount of information contained in the transition matrix of a bacterial genome beyond what these studies were looking for, and the existence of an optimal word length indicates that a lower-order Markov model can capture the majority of the information contained in higher-order models. The application of finite state Markov chain models to identify patterns that exist in bacterial genomes can help in understanding molecular change, in developing molecular criteria for classification, and in exploring the boundaries of what may (or may not) constitute a viable genome sequence.

Sequenced bacterial genomes span a size range of approximately two orders of magnitude, from *Carsonella ruddii* (∼0.15 MB) (*Nakabachi et al., 2006*) to *Sorangium cellulosum* (∼13 MB) (*Schneiker et al., 2007*), and a range of %G + C content from a low of ∼17% in *Carsonella ruddii* to ∼75% in *Anaeromyxobacter dehalogenans* (*Sanford, Cole & Tiedje, 2002*). If we consider the set of all possible bacterial chromosomes to include every closed circular DNA sequence that fits within these ranges, the number of distinct chromosomal sequences would be overwhelming. Determining the subset of probable bacterial chromosomes from the set of possible bacterial chromosomes is a problem whose complexity is analogous to protein structure prediction. To begin addressing this problem, we can apply heuristics based on biological phenomena considered to be ubiquitous. For example, we might propose that a sequence must contain codons, open reading frames, regulatory sequences, and a certain set of "essential" genes in order for it to be included in the probable subset. Applying these kinds of heuristics renders the subset of probable chromosomes much smaller that the set of possible chromosomes, but it would still be an overwhelmingly large number. Also, the boundaries of the subset would not be hard, since consensus on parameters such as the number of open reading frames and the list of essential genes would be impossible.

An independent and complementary approach to developing heuristics to limit the subset of probable bacterial chromosomes would be to base them on sequence patterns identified either as ubiquitous or extremely rare. This type of heuristic would not rely on a biological interpretation of sequence data, but rather on definable sequence patterns that are highly likely or unlikely to occur in the population based on their appearance within a representative sub-population. A few heuristics have already been proposed. For example, Lawrence and Ochman summarized four salient features of prokaryotic genomes (*Lawrence & Ochman, 1997*): (1) base composition varies widely among bacterial species; (2) base composition is related to phylogeny; (3) base composition is relatively homogeneous over the entire bacterial chromosome; and (4) within each species, the first, second, and third positions of codons, as well as the genes for structural RNAs, have characteristic base compositions. Once defined, these features can be explored and parameterized into models capturing certain properties.

Despite limited available data, early studies made some very important observations. Kariin et al. identified correlations between neighboring nucleotides (*Kariin & Burge, 1995*) (i.e., the probability of appearance of the $n$th nucleotide depends on the $n - 1$ nucleotides), and concluded that dinucleotide frequencies carry a phylogenic signal. Goldman and others discovered that tri- and tetranucleotide correlations exist in bacterial sequences (*Goldman, 1993*; *Karlin, Campbell & Mrázek, 1998*; *Karlin, Mrázek & Campbell, 1997*). Tetranucleotide frequencies have also been found to carry a phylogenetic signal, and to reflect high-order information beyond third codon biases that are not present in the analysis of single genes (*Pride et al., 2003*). The study by *Pride et al. (2003)* looked at tetranucleotide usage conservation in 27 microbial genomes, and compared a tree based on tetranucleotide usage departures to that of 16S rRNA trees. They concluded that tetraucleotide usage patterns are conserved by taxa, and that usage departure is a measure

of how far tetranucleodide frequencies diverge from the expectations under a null-model, which in their case was designed to remove any sequence bias. This approach has been useful in identifying under- and over-represented oligonucleotides (*Almagor, 1983*; *Karlin, Mrázek & Campbell, 1997*; *Schbath, Prum & De Turckheim, 1995*).

## MATERIALS AND METHODS

The complete DNA sequences and 16S ribosomal DNA sequences were collected for 906 closed bacteria from GenBank (*Benson et al., 2004*) (for a complete list see Text S1). For organisms having multiple chromosomes, the major chromosome was selected as representative of the genomic sequences of the respective organism. Our analysis indicates that the DNA sequence of the major chromosome in bacteria has similar statistical properties in regards to nucleotide probabilities as a sequence constructed by appending all the chromosomes for that organism, excluding plasmids (data not shown). All software developed for this work was written in C++, except where otherwise noted.

### Constructing the 16S rRNA tree

The ribosomal DNA sequences for each of the 906 bacteria were obtained from GenBank, and the DNA sequence corresponding to the 16S ribosome was written to a single FASTA file. In organisms having multiple copies of 16S rRNA, the first copy relative to the $5'$ direction was chosen as representative of the organism (*Acinas et al., 2004*). The 16S rRNA sequences were aligned using MUSCLE (*Edgar, 2004*). Aligned 16S rRNA sequences were bootstrapped with 100 replicates and transformed into distances using the F84 (*Kishino & Hasegawa, 1989*) model available in the Phylip package (*Felsenstein, 2005*). We chose to use the F84 distance method because, unlike other methods (e.g., Jukes and Cantor's (*Jukes & Cantor, 1969*) and K80 (*Kimura, 1980*)), it allows for both unequal base frequencies and unequal transition/transversion probabilities. The base frequencies and transition/transversion probabilities are estimated from the data, and the distances can be interpreted as a maximum likelihood estimate of the divergence time; this provides an accurate representation of bacterial sequence dynamics. Each replicated distance matrix was clustered using the Neighbor-joining method (*Saitou & Nei, 1987*). Neighbor-joining was used because of its speed and accuracy when given a correct distance matrix. A majority-rule consensus tree was calculated using Phylip (Phylip formatted tree available as Text S2). Tree visualizations shown in this paper were produced using Dendroscope (*Huson et al., 2007*) and ladderized right.

### Constructing the transition tree

The frequency of each genomic subsequence and its reverse complement ($3' \rightarrow 5'$) of length $n$ appearing in each bacterial genome was explicitly counted. The transition probabilities were estimated for the $k$th-order transition matrix ($k = n - 1$), for $0 \leq k \leq 5$, from the subsequence frequencies. The Euclidean distance was computed between each transition matrix describing each of the 906 bacterial sequences for a given order of Markov chain model. The Euclidean distances were clustered using the Neighbor-joining

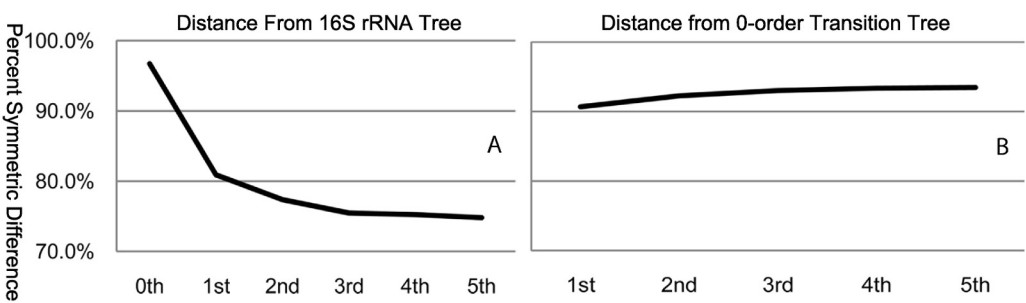

**Figure 1** **Percent symmetric difference of each order transition tree relative to the 16S rRNA tree (A) and the zero-order transition tree (B).** The greatest change in symmetric difference between the 16S rRNA tree and the tree based on transition matrices occurs between 0th order and 3rd order, with only a very small change thereafter. Similarly, the greatest symmetric difference between the 0th order transition tree and higher-order trees becomes relatively asymptotic after the 3rd order.

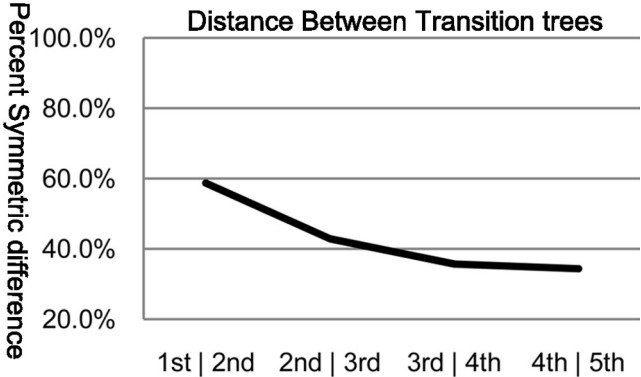

**Figure 2** **Percent symmetric difference between subsequent orders of transition trees.** The symmetric difference between the subsequent order transition trees becomes relatively asymptotic after the 3rd|4th order.

method available in the Phylip package (Phylip formatted tree available as Text S3). Tree visualizations were produced using Dendroscope (*Huson et al., 2007*) and ladderized right.

## Determination of tree similarity

A direct method of assessing tree similarity comes from set theory and is referred to as the symmetric difference (*Robinson & Foulds, 1981*). The symmetric difference of a tree structure is the total number of partitions that differ between the two trees. We used the percent symmetric difference, which is the symmetric difference ($D_s$) divided by the maximum symmetric difference ($D_{max}$), with $D_{max} \approx 2n - 6$ for $n$-number of taxa. The significance of $D_s$ for a given number of taxa can be estimated empirically, and is shown to be asymptotic, with a convergence rate dependent on $n$ (*Steel & Penny, 1993*). For $n = 30$, any $D_s < (D_{max} - 2)$ is significant, with $p < 0.01$. The symmetric difference method as implemented in the Phylip package was used for the data presented in Figs. 1 and 2.

**Peer**J

## Markov models of bacterial chromosomes

A chromosome sequence can be modeled as a finite state space Markov chain, with each of the four nucleotides (A, T, G, C) represented by a single state with transition probabilities $P_A$, $P_T$, $P_G$ and $P_C$ respectively. This representation is memoryless, in that the appearance of any nucleotide at any position is completely independent of any other. This is also referred to as a 0th order Markov model, and in this context can only capture biases in the relative frequency of appearance of the nucleotides (e.g., G + C bias and A/T fraction bias). The transition matrix, $\theta$, for the 0th order Markov model describing the finite state space Markov chain is:

$$\theta = [P_A, P_T, P_G, P_C].$$

In higher-order Markov models, transition probabilities are conditional on the previous $k$ bases (for $k > 0$). For example, we can consider a 1st order Markov model with transition probabilities $P_{A|A}$, $P_{T|A}$, $P_{G|A}$, $P_{C|A} \cdots P_{G|C}$, $P_{C|C}$, where $P_{i|j}$ is the probability of the $i$th nucleotide following the $j$th nucleotide. We can easily generalize this to describe the transition matrix for a $k$-order Markov model representing a genomic sequence, with $\theta = [i,j]4^k \times 4^1$.

## TESTING FOR MARKOVITY

We adapted an existing framework that is rooted in information theory to estimate the significance of a Markov property of order $k$. The framework tests for contingency tables and calculates the value of the $X^2$ statistic (*Anderson & Goodman, 1957*; *Kullback, Kupperman & Ku, 1962*), which tests the null hypothesis that the sequence is a realization of a stationary Markov chain of order $k - 1$, against the alternative hypothesis that it is a realization of a $k$-order stationary Markov chain. The application of this method to DNA sequences is discussed by *Avery & Henderson (1999)*. For very long sequences, such as chromosomal DNA, the resulting $X^2$ statistic for lower-order Markov models is almost guaranteed to be significant ($p$-value near zero). However, what can be assessed is the change in value of the statistic as the order and degrees of freedom increases. In other words, for increasing order $k$, we can observe the change in uncertainty between $k$ and $k - 1$ order models in terms of the $X^2$ statistic, and look for asymptotic behavior in its value. A constant $X^2$ value becomes less significant as the degrees of freedom increase. When the statistic's value is asymptotic in the presence of increasing degrees of freedom, the static value will inevitably become insignificant. For these data in this paper, the value of the $X^2$ statistic is significant for all sequences with $k \leq 3$ (see S4 for a table of the statistics for each chromosome considered in this work) and nearly asymptotic at $k = 3$ for the majority. Any violations to the second observation are because of sequences with relatively short lengths, high G + C bias, or some combination of the two.

## RESULTS AND DISCUSSION

The goal of this work is not to devise a new or improved method of phylogenetic inference, or to imply that Markov models are superior to other methods. Rather, our goal is to

address the following three questions (1) is there a universal Markov property present in whole bacterial DNA sequences; (2) to what extent (order) does this property hold true; and (3) is the existence of the Markov property biologically relevant.

Using the complete nucleotide sequences of 906 bacteria, including its complement, and excluding plasmids and minor chromosomes (*Benson et al., 2004*) (see Text S1 for a complete list or organisms), we estimated the 0th–5th order transition matrices, describing the respective order Markov chain model for each. The 5th order model intersects at least two codons and, given the length of bacterial genomes, it is still short enough to allow sufficient oligonucleotide frequencies to avoid sparse transition matrices. We then calculated the Euclidean distance between each pair of transition matrices for each order model (one distance matrix for each order Markov chain model for all chromosomes) and produced a cladogram from the distances based on the Neighbor Joining method (*Saitou & Nei, 1987*). We refer to this kind of tree as a "transition tree".

Branching patterns of trees based on alignments of 16S ribosomal RNAs are an accepted method to represent phylogeny (*Fox et al., 1980*; *Woese & Fox, 1977*). To see if this is also a characteristic of the transition tree, we performed a comparison between each transition tree and a 16S rRNA tree constructed in similar fashion (see Methods for a detailed description). Briefly, 16S rRNA sequences for each of 906 bacteria were collected from GenBank and aligned against one another using MUSCLE (*Edgar, 2004*). Alignments were bootstrapped with replacement (*Felsenstein, 1985*), transformed into a distance matrix, clustered using the Neighbor Joining method, and a cladogram was produced for visual comparison.

## Comparisons between the 16S rRNA and transition tree topologies

Using the symmetric difference method (*Robinson & Foulds, 1981*) of comparing tree topologies, we calculated the percent symmetric difference of each transition tree (1st–5th order) relative to the 16S rRNA tree (Fig. 1B) and to the 0th order transition tree (Fig. 1A). Previous research on the distribution of $D_s$ from simulation data has shown it to be asymptotic in nature, with convergence dependent on the number of taxa. These findings are summarized in *Steel & Penny (1993)*, and suggests that for trees with more than a moderate number of taxa, any $D_s < D_{max}$ is significant, (e.g., for $n = 30$, any similarity of more than a few partitions is very unlikely). Therefore, no similarity in topology is predicted between randomly placed nodes in trees with a large number of taxa. As shown in Fig. 1A, the congruence, $(1 - (D_s/D_{max})) \times 100$, between the 0th order tree, which is a function of G + C content alone, and the 16S rRNA tree is low ($D_s/D_{max} \times 100 = 96.7\%$). However, as summarized by Steel and Penny, even this small difference from $D_{max}$ is significant, and this suggests that there is some influence of G + C content reflected in the 16S rRNA tree. A similar conclusion can be reached by examining Fig. 1B. The percent symmetric difference between the 0th order transition tree and the higher-order transition trees is large (90.7%–93.5%), but even this small degree of congruence is considered significant, and it reflects the influence of the 0th order model on the higher-order models.

Interestingly, the effects of G + C content are rather stable beyond the 2nd order model, in that the percent symmetric difference between the 0th order model and higher order models (beyond 2nd order) does not change by a large amount (93.0%–93.5%). These observations lead us to conclude that G + C content bias has a real but relatively small influence on both the 16S rRNA tree and the transition tree.

The symmetric difference between the 16S rRNA tree and each of the transition trees decreases most between the 0th and 3rd orders (96.7%–75.5%), with little additional decrease between the 3rd and 5th order (75.2%–74.8%). These data lead us to conclude the following: (1) the 3rd order transition tree shares ∼25% similarity to the 16S rRNA tree; (2) this congruence is greater than for trees based on lower-order models; (3) this congruence is similar to trees based on higher-order models. These conclusions are further supported by data presented in Fig. 2. We calculated the percent symmetric difference between subsequent orders of transition trees and observed that from 3rd until 5th order, each order transition tree shares approximately 65% of its partitions with its previous and subsequent order trees. This leads us to conclude that large decreases in symmetric difference between subsequent orders of transition trees stop after the 3rd order. Of course, if we continued to increase the order of the Markov models indefinitely, the subsequent tree topologies produced by $k-1$ and $k$ order models would eventually converge. This is due to the increasingly sparse transition matrices. For a given sequence, the transition matrix would approach the null set, with only two elements populated (that corresponding to $(1 \ldots n-1)$ and to $(2 \ldots n)$ for a sequence of length $n$) with a frequency count of 1. The resulting distance matrix, based on the sparse transition matrices, will reach steady-state. For any particular sequence, the complexity of the model necessary to achieve this convergence depends on many factors, including sequence length and G + C content bias. Convergence is inevitable, however, because it is inherent in the model. In other words, in the most extreme case, we can always find a model of order equal to the sequence length $n-1$.

## Data bias

Bias must be considered because it exists in the collection of sequenced bacteria. Some genera (e.g., *Escherichia, Streptococcus*, and *Bacillus*) are overrepresented, while others are underrepresented. We must therefore consider the possibility that the 16S rRNA tree and the transition tree show a greater degree of congruence in more closely related species, so that the overrepresented genera would inflate the overall congruence in topology between the transition trees and the 16S rRNA tree. To determine if this effect exists, four overrepresented genera, *Escherichia* (*Höhl & Ragan, 2007* species), *Streptococcus* (*Kullback, Kupperman & Ku, 1962* species), *Bacillus* (*Gelfand, Kozhukhin & Pevzner, 1992* species) and *Burkholderia* (*Fox et al., 1980* species), totaling 119 species (∼13% of the data collection) were chosen, and 16S rRNA and 3rd order transition trees were constructed. This subset of species was selected to represent an exaggerated sequencing bias so that, if the observed congruence between 16S rRNA trees and transition trees is partly due to this bias, it should be amplified in this subset. Instead, the symmetric difference between these

16S rRNA Tree

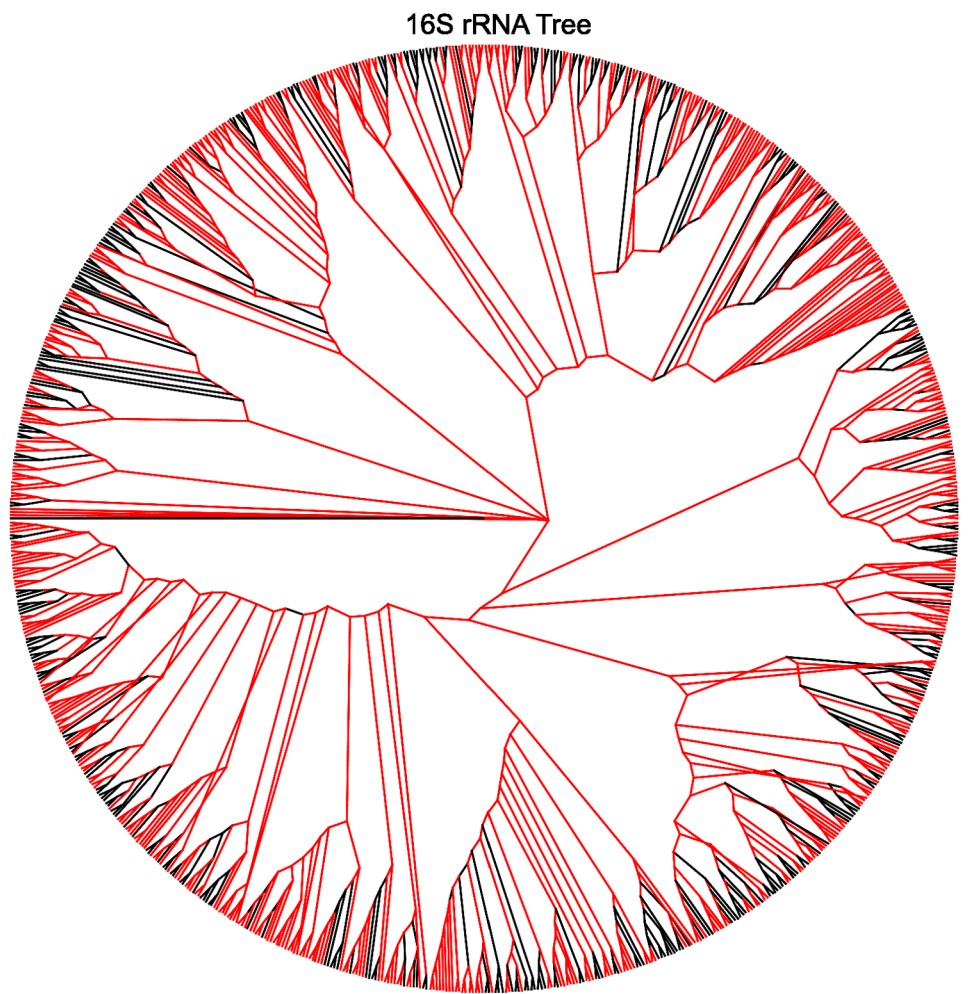

**Figure 3 The symmetric difference between the 16S rRNA tree and the third-order transition tree.** Branches marked in red represent disagreement in topology between the trees.

trees was calculated as 76.7%, which is very close to the 75.5% measured using the entire 906 bacteria. We therefore conclude that sequence bias has no significant impact on these results.

## Topology of 16S rRNA tree versus 3rd order transition tree

The symmetric difference between the 16S rRNA tree and the 3rd order transition tree is presented in Fig. 3 as the 16S rRNA tree, with branches in red representing disagreement between it and the 3rd order transition tree. The 16S rRNA tree and 3rd order transition tree from which Fig. 3 is derived are provided in supplementary materials (Text S2 and Text S3, respectively). In Figs. 4–6, the taxa of interest are shown in red, with the 16S rRNA tree on the left and the transition tree on the right. Comparisons are made relative to the transition tree, with all organisms of a particular genera of interest accounted for in both trees (as either a group member or outlier in the transition tree).

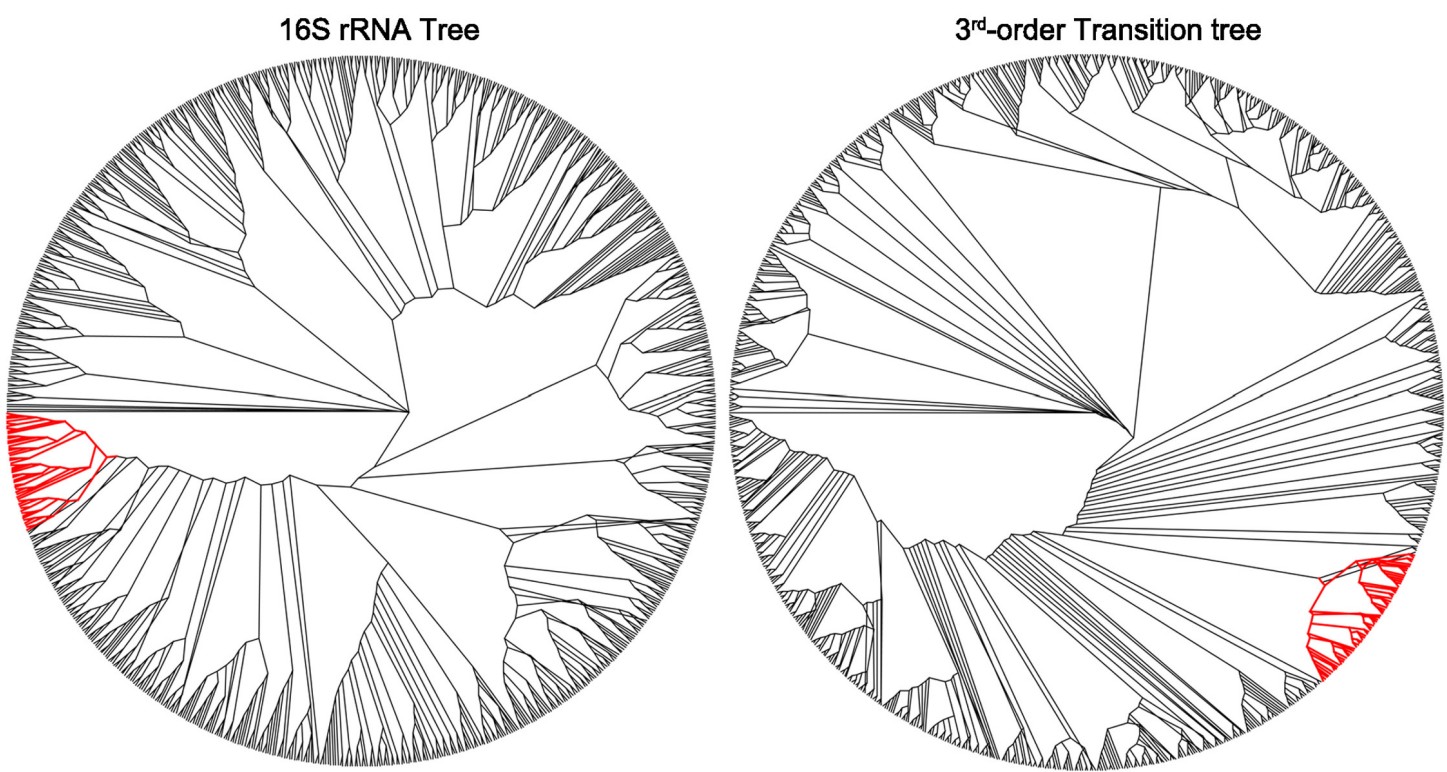

**Figure 4 A collection of Enterobacteriaceae consisting of *Salmonella*, *Escherichia* and *Shigella* as example of taxa which cluster similarly in the 16SrRNA and third-order transition trees.** The genus of interest appear in red in the radial cladogram. A list of the organisms is given, with species that are not included in the transition tree, but are included in the 16S rRNA tree in boldface type. *A.macleodii_Deep_ecotype, H.baltica_ATCC_49814, I.loihiensis_L2TR, K.koreensis_DSM_16069,* **L.acidophilus_NCFM,** *L.brevis_ATCC_367, L.casei_ATCC_334,* **L.delbrueckii_bulgaricus, L.delbrueckii_bulgaricus_ATCC_BAA-365, L.fermentum_IFO_3956, L.gasseri_ATCC_33323, L.helveticus_DPC_4571, L.johnsonii_FI9785, L.johnsonii_NCC_533,** *L.plantarum, L.plantarum_JDM1,* **L.reuteri_DSM_20016, L.reuteri_F275_Kitasato,** *L.rhamnosus_GG, L.rhamnosus_Lc_705,* **L.sakei_23K, L.salivarius_UCC118,** *Marinomonas_MWYL1, M._mobilis_JLW8, P.profundum_SS9, P.necessarius_asymbioticus_QLW_P1DMWA_1, P.necessarius_STIR1, P.atlantica_T6c, P.haloplanktis_TAC125, P.arcticum_273-4, P.cryohalolentis_K5, Psychrobacter_PRwf-1, S.degradans_2-40,* **S.amazonensis_SB2B,** *Shewanella_ANA-3, S.baltica_OS155, S.baltica_OS185, S.baltica_OS195, S.baltica_OS223, S.denitrificans_OS217, S.frigidimarina_NCIMB_400, S.halifaxensis_HAW_EB4,* **S.loihica_PV-4,** *Shewanella_MR-4, Shewanella_MR-7, S.oneidensis, S.pealeana_ATCC_700345, S.piezotolerans_WP3, S.putrefaciens_CN-32, S.sediminis_HAW-EB3, Shewanella_W3-18-1, S.woodyi_ATCC_51908, T.crunogena_XCL-2,* **T.denitrificans_ATCC_33889,** *V.cholerae, V.cholerae_M66_2, V.cholerae_MJ_1236, V.cholerae_O395, Vibrio_Ex25,* **V.fischeri_ES114,** *V.harveyi_ATCC_BAA-1116, V.parahaemolyticus, V.splendidus_LGP32, V.vulnificus_CMCP6, V.vulnificus_YJ016, Y.enterocolitica_8081, Y.pestis_Angola, Y.pestis_Antiqua, Y.pestis_biovar_Microtus_91001, Y.pestis_CO92, Y.pestis_Nepal516, Y.pestis_Pestoides_F, Y.pseudotuberculosis_IP_31758, Y.pseudotuberculosis_IP32953, Y.pseudotuberculosis_PB1, Y.pseudotuberculosis_YPIII.*

There is good agreement between the 16S rRNA tree and the 3rd order transition tree in several places; Fig. 4 presents a large collection of Enterobacteriaceae as an example. This grouping includes *Salmonella*, *Escherichia* and *Shigella*, and the transition tree shows consistent grouping of each genus as compared to the 16S rRNA tree. The 16S rRNA sequences of *Shigella* and *Escherichia* are very homologous, and this results in some species from each genus being shuffled within the 16S rRNA tree as opposed to the transition tree, which is more sensitive to sequence bias. This shuffling is not observed in the transition tree.

Figure 5 illustrates differences in how the genus *Streptococcus* clusters in the 16S rRNA tree versus the transition tree. In the 16S rRNA tree, all of the Streptococci form one cluster,

**Peer**J

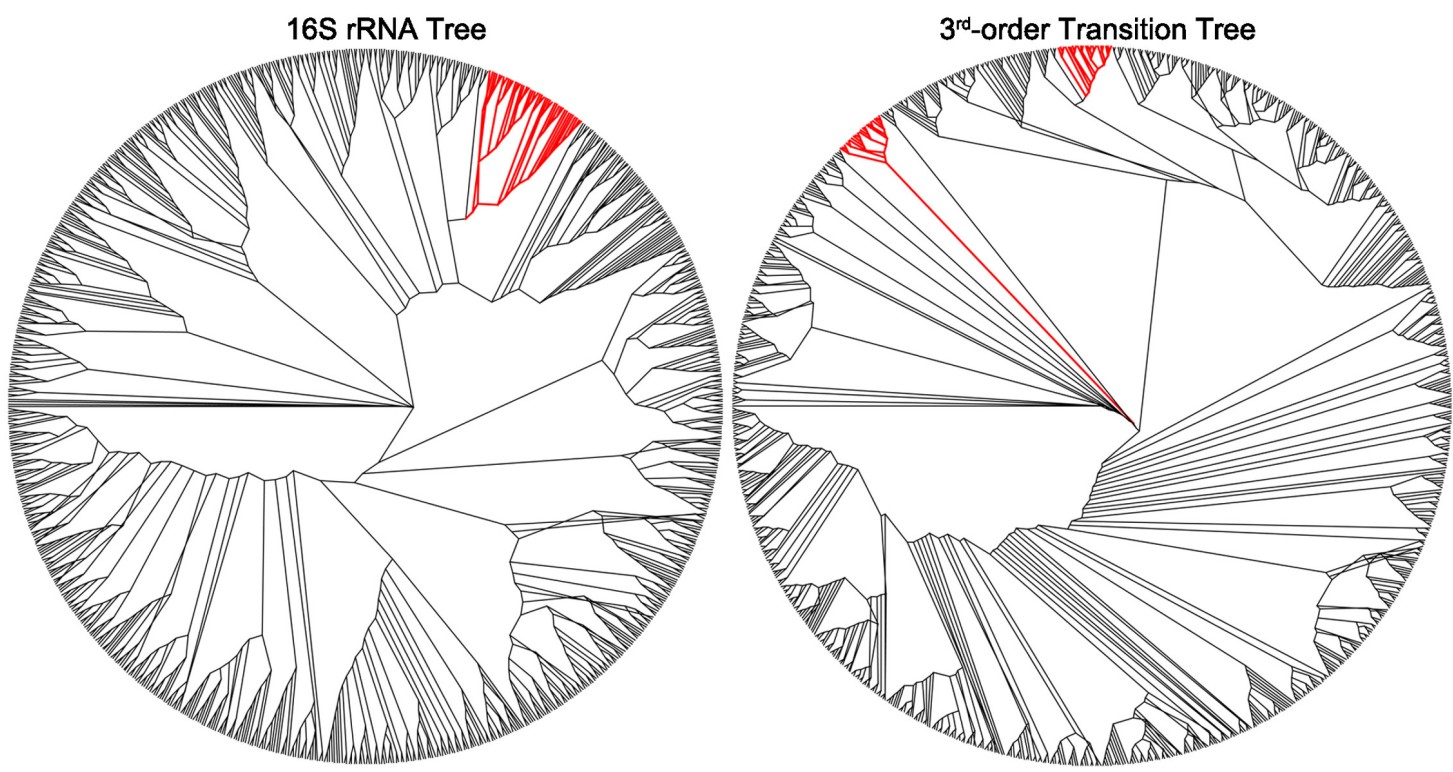

**Figure 5 Genus Streptococcus appear in two distinct clusters in the third-order transition tree, but are assigned one cluster in the 16SrRNA tree.** The genus of interest appears in red in the radial cladogram. A list of the organisms is given. **Group 1**: *S.equi_4047, S.equi_zooepidemicus, S.equi_zooepidemicus_MGCS10565, S.gordonii_Challis_substr_CH1, S.sanguinis_SK36, S.pneumoniae_70585, S.pneumoniae_JJA, S.pneumoniae_D39, S.pneumoniae_R6, S.pneumoniae_P1031, S.pneumoniae_G54, S.pneumoniae_Taiwan19F_14, S.pneumoniae_ATCC_700669, S.pneumoniae_CGSP14, S.pneumoniae_Hungary19A_6, S.pneumoniae_TIGR4, S.suis_05ZYH33, S.suis_98HAH33, S.suis_SC84, S.suis_P1_7, S.suis_BM407* **Group 2**: *S. agalactiae_2603, S.agalactiae_NEM316, S.agalactiae_A909, S.dysgalactiae_equisimilis_GGS_124, S.pyogenes_M1_GAS, S.pyogenes_MGAS9429, S.pyogenes_MGAS10270, S.pyogenes_NZ131, S.pyogenes_MGAS10750, S.pyogenes_MGAS10394, S.pyogenes_MGAS8232, S.pyogenes_MGAS315, S.pyogenes_MGAS5005, S.pyogenes_MGAS6180, S.pyogenes_MGAS2096, S.pyogenes_Manfredo, S.pyogenes_SSI-1, S.thermophilus_CNRZ1066, S.thermophilus_LMG-18311, S.thermophilus_LMD-9, S.uberis_0140J, S.mutans.*

whereas in the transition tree there are two separate clusters. The two clusters do not divide based on hemolytic properties, serogroup or habitat, however, each group has a distinct G + C content ($p < 0.05$ with Students t-test) with group one $\mu = 40.43\%$, $\sigma^2 = 1.05\%$, $n = 21$ and group two $\mu = 37.92\%$, $\sigma^2 = 1.43\%$, $n = 22$, where $\mu$ is the mean G + C content, $\sigma^2$ is the variance about the mean, and $n$ is the number of samples. There is a distinct difference in nucleic acid content between the two groups of Streptococci that does not appear to follow the typical physiological traits used to define these organisms. In this case, the transition tree is detecting clear molecular differences between otherwise similar organisms.

Figure 6 highlights a group of bacteria that cluster tightly in the transition tree (with outliers in boldface type), but are separated into distinct groups in the 16S rRNA tree. This group includes members of the *Polynucleobacter, Psychrobacter, Marinomonas, Shewanella* and *Vibrio* genera, with a G + C content range of approximately 40%–49%. Most of these organisms are associated with cold-water aquatic habitats. It is known that thermophiles

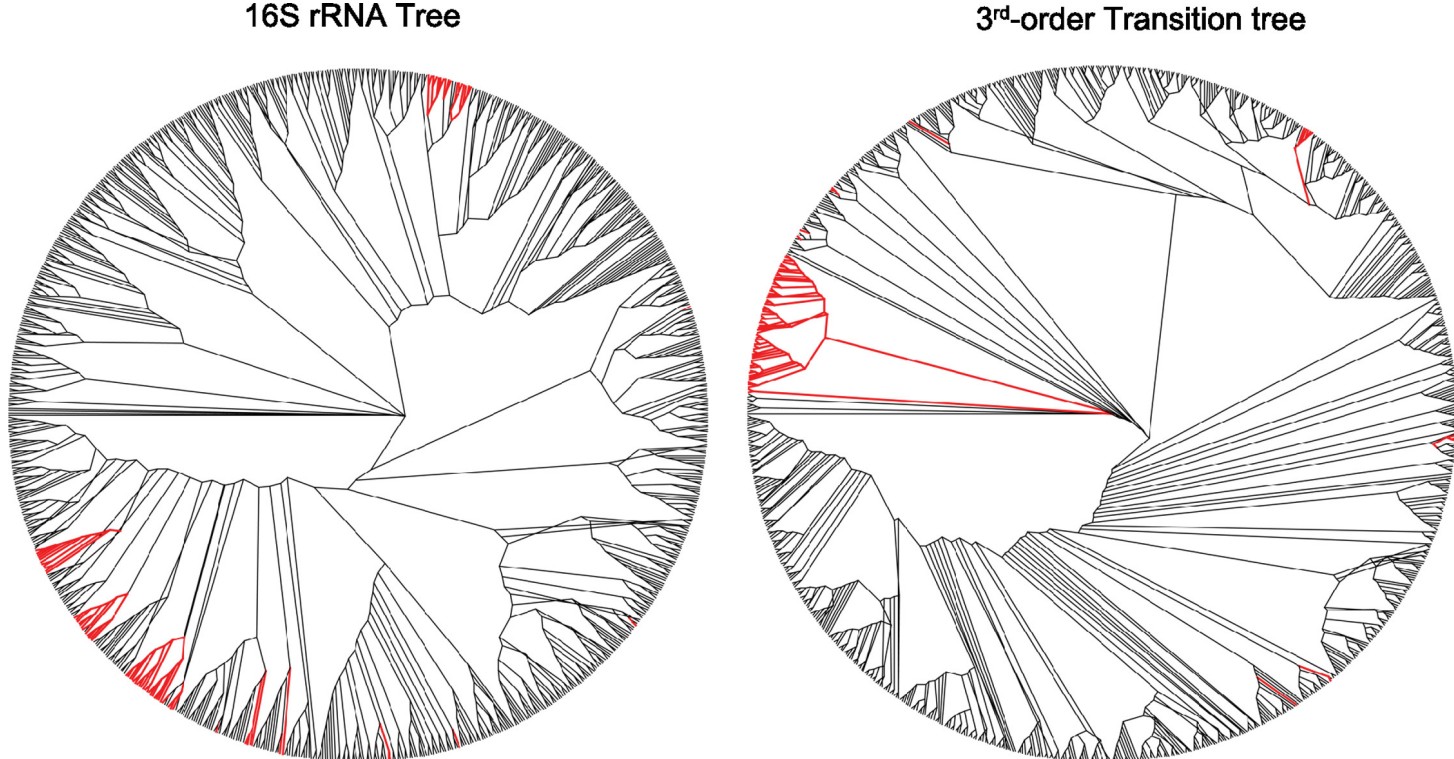

**Figure 6 A group of mostly aquatic bacteria that cluster together in the third-order transition tree, but are dispersed in the 16S rRNA tree.** The genus of interest appear in red in the radial cladogram. A list of the organisms is given with those that appear outside the cluster in the transition tree in boldface type. *Shewanella_sediminis_HAW-EB3, Shewanella_woodyi_ATCC_51908, Alteromonas_macleodii_Deep_ecotype_, Saccharophagus_degradans_2-40, Pseudoalteromonas_haloplanktis_TAC125, Methylotenera_mobilis_JLW8, Psychrobacter_arcticum_273-4, Psychrobacter_cryohalolentis_K5, Psychrobacter_PRwf-1, Pseudoalteromonas_atlantica_T6c, Shewanella_ANA-3, Shewanella_MR-4, Shewanella_MR-7, Shewanella_baltica_OS155, Shewanella_baltica_OS185, Shewanella_baltica_OS195, Shewanella_baltica_OS223, Shewanella_oneidensis, Shewanella_putrefaciens_CN-32, Shewanella_W3-18-1, Shewanella_denitrificans_OS217, Shewanella_halifaxensis_HAW_EB4, Shewanella_pealeana_ATCC_700345, Shewanella_piezotolerans_WP3, Shewanella_frigidimarina_NCIMB_400, Photobacterium_profundum_SS9, Vibrio_cholerae, Vibrio_cholerae_M66_2, Vibrio_cholerae_O395, Vibrio_cholerae_MJ_1236, Vibrio_vulnificus_CMCP6, Vibrio_vulnificus_YJ016, Vibrio_Ex25, Vibrio_harveyi_ATCC_BAA-1116, Vibrio_parahaemolyticus, Vibrio_splendidus_LGP32, Marinomonas_MWYL1, Hirschia_baltica_ATCC_49814, Polynucleobacter_necessarius_asymbioticus_QLW_P1DMWA_1, Polynucleobacter_necessarius_STIR1, Idiomarina_loihiensis_L2TR, Yersinia_enterocolitica_8081, Yersinia_pestis_Angola, Yersinia_pestis_Nepal516, Yersinia_pestis_Antiqua, Yersinia_pestis_biovar_Microtus_91001, Yersinia_pestis_CO92, Yersinia_pseudotuberculosis_IP32953, Yersinia_pseudotuberculosis_PB1_, Yersinia_pseudotuberculosis_IP_31758, Yersinia_pseudotuberculosis_YPIII, Yersinia_pestis_Pestoides_F, Lactobacillus_brevis_ATCC_367, Lactobacillus_plantarum, Lactobacillus_plantarum_JDM1, Lactobacillus_casei_ATCC_334, Lactobacillus_rhamnosus_GG, Lactobacillus_rhamnosus_Lc_705, Kangiella_koreensis_DSM_16069, Thiomicrospira_crunogena_XCL-2,* **Vibrio_fischeri_ES114, Lactobacillus_sakei_23K, Lactobacillus_reuteri_DSM_20016, Shewanella_amazonensis_SB2B, Shewanella_loihica_PV-4, Lactobacillus_delbrueckii_bulgaricus, Thiomicrospira_denitrificans_ATCC_33889, Lactobacillus_acidophilus_NCFM**.

exhibit preferences in the first codon for G + C, due to the higher melting temperatures (*Kreil & Ouzounis, 2001*; *Tekaia, Yeramian & Dujon, 2002*); the converse of this is a similarly reasonable explanation. Thermophobic bacteria may prefer A + T at the first codon due to lower separation energies required during replication. Although members of Yersinia and six species of Lactobacillus may initially appear to contradict this observation, this may not be the case. *Yersinia pseudotuberculosis* is a soil- and waterborne human pathogen, and the closest known ancestor of *Yersinia pestis* (*Achtman et al., 1999*), and many species of Lactobacillus can be found in marine sediment. There is further evidence

in support of our aquatic hypothesis within the other genera. Two species of *Shewanella* are located outside the cluster, *S. amazonensis* and *S. loihica*. Both of these organisms are psychrophobic, whereas the *Shewanella* species within the cluster grow well at low to moderate temperatures. Also, *Vibrio* is a genus of proteobacteria that are a common cause of food-borne illness resulting from infected seafood. *V. fischeri*, which is the only Vibrio outlier, is unique among *Vibrio* species because it is apathogenic and found predominantly in symbiosis with various marine animals. We hypothesize that these species represent outliers on the transition tree because they occupy different habitats from their 16S nearest neighbors.

Habitat has been shown to influence genomic composition (*Foerstner et al., 2005*). Perhaps the difference between the 16S rRNA tree and 3rd order transition tree illustrated in Fig. 4 is an example of that influence.

## CONCLUSIONS

These data and analyses lead us to the following three observations: (1) in nearly all bacterial chromosomes there is a significant long-range nucleotide correlation that extends beyond the 2nd order; (2) similarity trees constructed on matrices derived from these correlations have a statistically significant overlap with 16S RNA trees and, when divergent, may reveal functional differences between species; (3) the apparent ubiquity of these correlations may place practical limitations on what will or will not evolve to become a bacterium.

These observations cannot easily be explained by our understanding of biology. Overall G + C bias is a 0th order property, so that its influence is completely defined by the independent probabilities of each of the four nucleotides. A codon is three nucleotides long, so codon bias within open reading frames is a 1st order (binucleotide) or 2nd order (trinucleotide) correlation. Any correlations that extend beyond 2nd order reflect a mechanism or mechanisms that drive the nucleic acid order beyond the length of a codon.

We have also shown that the transition matrices for a large number of chromosomes exhibit a phylogenetic correlation. From the matrices we can build transition trees that are statistically similar to 16S rRNA trees, and we propose that some of the differences between transition and 16S trees may be due to influences from ecological niche and/or habitat. Proximity would present organisms that occupy similar habitats, such as cold water, with the opportunity to share genetic material that increases their likelihood for survival, such as anti-freeze genes (*Gilbert et al., 2004*). Although transfer of small bits of genetic material would not account for similarity of transition matrices between whole chromosomes of distantly related organisms, DNA sharing has been previously observed on a much larger scale. Specialized bacteria that occupy the same habitat or ecological niche also may experience convergent evolution (*Audic et al., 2007*; *Suen, Goldman & Welch, 2007*). Horizontal gene transfer (HGT) is known to play a major role in how bacteria acquire new genetic material. It seems logical that organisms within the same habitat might acquire similar genomic characteristics via HGT.

The Markov property we have described appears to be ubiquitous. We were able to identify the property in all of the 906 chromosomes we studied, and it has been estimated that there are $\sim 10^8$ bacterial species on the Earth (*Curtis, Sloan & Scannell, 2002*; *Schloss & Handelsman, 2004*). Using the statistical "rule of three", we can be 95% confident that the rate of this phenomenon is no less frequent than 301 in 302 bacterial chromosomes. We therefore conclude that the majority of all of the bacterial species will have this Markov property in their chromosomes, and this likely represents a statistical heuristic that limits the sequence space of probable bacterial chromosomes.

What, if any effect this conclusion has on our ability to select the probable from the much larger number of possible bacterial chromosomes is impossible to quantify, but we may be able to illustrate some of its impact and provide context through example. If any one of the set of possible bacterial chromosomes can be represented as a random closed circle of nucleotides, and if we assume one biology-based heuristic –that it can be any integral length between the smallest (0.15 Mb) and the longest (15 Mb) sequenced chromosome, then there are $\sum_{n=\alpha}^{\beta} 4^n = \frac{4}{3}(4^\beta - 4^{\alpha-1}) \approx 10^{9,000,000}$ possible bacterial chromosomes ($\alpha = 0.15$ M and $\beta = 15$ M) each with an equal probability of occurrence. If we now consider that most bacterial chromosomes have a compositional bias (e.g., G + C content), some of the possible combinations become more or less probable. If we then consider higher-order compositional biases, say an A/T fractional bias in addition to a G + C content bias, then we can be even more specific about probable and improbable chromosomes.

With the existence of a high-order Markov process, the number of variables (states) increases exponentially with each increase in model order. This allows a more precise determination of the probability of a particular sequence (i.e., greater resolution of transition probabilities), and thereby the identification of more sequences that are unlikely to be bacterial chromosomes. Let $X_L^K$ define a sequence of $K$ letters over an alphabet of $L$ characters, then the probability of sequence $X_L^K$ is: $P(x_L^K) = \prod_{j=1}^{K} P(X_j = x_j | X_L^{j-L} = x_L^{j-L})$, where $X_j$ represents the nucleotide at position $j$ with $x_j$ as its realization. For a DNA sequence (and assuming a 3rd-order Markov Model), $L = K = 4$. In the trivial case, where each character (nucleotide) is equally likely to occur, it can be easily shown that $P(x_L^K) = \frac{1}{L^K}$ and the expected frequency $f(x_L^K) = \frac{N-K-1}{L^K} \approx \frac{N}{L^K}$ for $K \ll N$. For any sequence that is the result of a 3rd-order Markov process and modeled as such, we get $L^K = 4^4$ times more states than with a 0-order model. In other words, we get 256 times greater resolution of transition probabilities than if we just consider limitations of G + C bias and chromosome length.

We know that many of the biological constraints placed on an organism limit the number of possible combinations that can result in a viable genomic sequence, but these constraints seem difficult to quantify. Now that we have a significant sample size of sequenced bacterial chromosomes, we can identify some of the constraints through statistical methods, and perhaps also uncover new biological phenomena.

## ACKNOWLEDGEMENTS

The authors would like to thank William T. Starmer, Department of Biology, Syracuse University, for his guidance and thoughtful suggestions during the course of this work and for his critical review of this manuscript. We also want to express our appreciation to Barry Goldman, Monsanto Corp., for his insights regarding the possible data bias and into the lifestyle of many of the 906 bacteria included in this study and his time spent reading the drafts. We would also like to thank Laura Welch for her editorial comments.

### Funding

This work was supported by the Syracuse University College of Arts and Sciences and the National Science Foundation's CAREER Award (MCB-0746066 to RDW) and EFRI Award (EFRI-1137186 to RDW). The funders had no role in study design, data collection and analysis, decision to publish, or preparation of the manuscript.

### Grant Disclosures

The following grant information was disclosed by the authors:
National Science Foundation's CAREER Award: MCB-0746066.
EFRI Award: EFRI-1137186.

### Competing Interests

There are no competing interests.

### Author Contributions

- Aaron D. Skewes conceived the original idea and designed the experiments, performed the experiments, analyzed the data, wrote the paper.
- Roy D. Welch conceived the original idea, contributed to the design of experiments, contributed reagents/materials/analysis tools, wrote the paper.

### Supplemental Information

Supplemental information for this article can be found online at http://dx.doi.org/10.7717/peerj.127.

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
