# Peer review of "A Markovian analysis of bacterial genome sequence constraints"

_PeerJ, doi:10.7717/peerj.127_

## Round 0.1 · original submission · Major Revisions

As you will see below, the two reviewers have substantially different opinions on the manuscript. Because of this discrepancy, I read the manuscript in detail myself. I must say that I share many of the concerns voiced by referee #2 and therefore ask that you carefully address them. In addition, please address the following points:

Most importantly, I am concerned that the work is not well embedded in the recent literature on the topic. To my understanding, the k-order Markov models you use are equivalent to (k+1)-mer word frequency counting---the representation most commonly used in related previous work. Yet many such relevant studies that are not covered. In particular, there has been previous work identifying the optimal Markov order for phylogenetic inference and numerous results identifying biases beyond trinucleotides.

As starting point, here are a few highly relevant studies:

Höhl M, Ragan MA (2007) Is multiple-sequence alignment required for accurate inference of phylogeny? Syst Biol 56: 206–221. doi:10.1080/10635150701294741.

Jun S-R, Sims GE, Wu GA, Kim S-H (2010) Whole-proteome phylogeny of prokaryotes by feature frequency profiles: An alignment-free method with optimal feature resolution. Proceedings of the National Academy of Sciences 107: 133–138. doi:10.1073/pnas.0913033107.

Dai Q, Wang T (2008) Comparison study on k-word statistical measures for protein: From sequence to “sequence space.” BMC Bioinformatics 9: 394. doi:10.1186/1471-2105-9-394.

Davenport CF, Tümmler B (2010) Abundant oligonucleotides common to most bacteria. PLoS ONE 5: e9841. doi:10.1371/journal.pone.0009841.

Elhai J (2001) Determination of bias in the relative abundance of oligonucleotides in DNA sequences. J Comput Biol 8: 151–175. doi:10.1089/106652701300312922.

Pascal G, Médigue C, Danchin A (2006) Persistent biases in the amino acid composition of prokaryotic proteins. Bioessays 28: 726–738. doi:10.1002/bies.20431.

On page 11 "similarity trees constructed on matrices derived from these correlations are in good agreement with 16S RNA trees". Given that the agreement is at best for ~25% of the split, this is an overstatement.

Typos:
- p4: tetrenucleotide
- p11: smaples

Reviewer 1 ·

Basic reporting

No Comments

Experimental design

No Comments

Validity of the findings

No Comments

Additional comments

The paper performs a large-scale Markovian analysis of 906 bacterial genome sequences. The main conclusion is that the genome sequences exhibit Markov property beyond the second-order, which places significant constraints on probable bacterial nucleotide sequences.

Overall, the technique employed is sound and the conclusion is valid. While the study of sequence comparison in bacteria is not new and the conclusion is not surprising, the few results on the differences between the tree constructed from 16S rRNA sequences and the transition tree constructed from whole genome sequences are interesting.

Reviewer 2 ·

Basic reporting

In general the manuscript is readable but there are some problems with clarity. For example
L147 "accepted trace of phylogeny" - What is a "trace of phylogeny" is it not just the accepted phylogeny?
L75 "and for a review (28)" - this does not make sense in the rest of the sentence

Further some things have not been fully described, for example sequence length for the bacterial sequences is not given.

There are also a few typos
L86 "bacterial chromosomes in most" should be "bacterial chromosomes is most"
L213 "the tree 16s RNA tree" should be "16s RNA tree"
Conclusions "These observations cannot easily be explained by our understanding biology" should be "These observations cannot easily be explained by our understanding of biology"

Experimental design

It is not clear why Markov Chains of different orders have been chosen to model this data. The previous study by Pride uses a 0th order model built on tetramers. This is basically a k-mer counting approach which has also been used previously for phylogenetic tree building. There is no explanation in the manuscript why the different order Markov Chain models would be better than k-mer counting for different k values. As the different order Markov Chains shown here have not been shown to be used elsewhere or shown to be better than previous methods it would be useful to show simulations exhibiting that this is a viable method, and indeed that it is better than a k-mer counting method (or citations which show this). It is also not shown whether the sequence lengths of the bacterial chromosomes (which are not given) have an effect on the accuracy of the model, and then a further effect on the tree.

It is not clearly explained why if the order of the Markov models was increased indefinitely then the subsequent tree topologies would eventually converge. And equally it is not clear what they would converge to. I would have thought that increasing the order of the Markov models will lead to sparse transition matrices (as suggested on line 142) which may make building trees from them difficult.

The methods used for building the 16S rRNA tree have not been justified, for example why has the F84 distance measure been chosen. I assume neighbor-joining was used due to the large number of sequences but this has not been stated. As it is known that different bacteria have different GC contents then it may make sense to use a phylogenetic method that takes this, or other known features, into account.

Validity of the findings

There are two big claims in the manuscript. Firstly that "the existence of a third order Markov Process in bacterial chromosomes is most likely universal", and secondly that "transition matrix usage is conserved in taxa". The first claim does not seem to be justified by the data in the manuscript. The fact that a 3rd order model gives a transition tree a lot closer to the phylogenetic tree than a lower order model, and that then increasing the order further does not make greater improvements, does not show that "in nearly all bacterial chromosomes there is a significant long-range nucleotide correlation that extends beyond the 2nd order". This shows as a whole there is evidence of 3rd order effects, but there is no way of knowing that all bacterial chromosomes show this effect. The manuscript later acknowledges that testing Markovidity in this data is very difficult but they have done some statistical tests to show Markovidity. They have however not shown any data for this. As the Markov property is the crux of all their claims much more evidence needs to be shown to prove they have indeed found it. Also the fact that proving Markovidity is difficult should be acknowledged when large claims are made and not only mentioned later on.

The second claim that "transition matrix usage is conserved in taxa" appears to have no evidence supporting it.

A number of the biological explanations for differences between the 16S rRNA phylogeny and the 3rd order transition tree are unconvincing. Why does the fact that Shigella and Escheria sequences are homologous explain the fact that they are shuffled in 16s rRNA tree and not the transition tree. Also why does the nucleic acid content between two groups of Streptococci show up in a 3rd order tree and not in the phylogeny. Dos this indicate that the method used to build the phylogeny could be improved to take this into account?

---

## Round 0.2 · Minor Revisions

Thank you for your revisions. There is only one outstanding concern—the need to clarify the difference between your k-order markov chain vs. (k+1)-mer counting. Surely this is something you can address in a minor revision of the submission.

Reviewer 1 ·

Basic reporting

No Comments

Experimental design

Although there is now a lot more background, I still feel that within the manuscript it is not completely clear why these models have been used instead of k-mer counting, and feel that a few sentences explaining this would be very useful.

Validity of the findings

No Comments

---

## Round 0.3 · accepted · Accept

Thank you for your prompt revision, which has cleared the last hurdle.